# Insights into Sensing of Murine Retroviruses

**DOI:** 10.3390/v12080836

**Published:** 2020-07-31

**Authors:** Eileen A. Moran, Susan R. Ross

**Affiliations:** Department of Microbiology and Immunology, University of Illinois at Chicago College of Medicine, 835 S. Wolcott Avenue, Chicago, IL 60612, USA; emoran9@uic.edu

**Keywords:** murine leukemia virus, mouse mammary tumor virus, PRR, PAMP, ALR, TLR, nucleic acid sensor

## Abstract

Retroviruses are major causes of disease in animals and human. Better understanding of the initial host immune response to these viruses could provide insight into how to limit infection. Mouse retroviruses that are endemic in their hosts provide an important genetic tool to dissect the different arms of the innate immune system that recognize retroviruses as foreign. Here, we review what is known about the major branches of the innate immune system that respond to mouse retrovirus infection, Toll-like receptors and nucleic acid sensors, and discuss the importance of these responses in activating adaptive immunity and controlling infection.

## 1. Introduction

The mammalian innate immune system serves as one of the first lines of defense against pathogens. It is composed of several receptors and signaling pathways involved in detecting foreign proteins, lipids, nucleic acid sequences or structures, and other components of pathogens, termed pathogen-associated molecular patterns (PAMPs). Pattern recognition receptors (PRRs) detect PAMPs and ensure that host cells initiate a targeted response that ultimately rids the organism of harmful pathogens. PRRs include, among others, Toll-like receptors (TLRs), retinoic acid-inducible gene-I (RIG-I)-like receptors (RLRs), absent in melanoma 2 (AIM2)-like receptors (ALRs), cyclic GMP-AMP synthase (cGAS), and members of the DEAD box (Asp-Glu-Ala-Asp/His) helicase and zinc finger-containing families. Upon PAMP detection, PRRs initiate signaling cascades that induce expression of type I interferons (IFNs), proinflammatory cytokines and chemokines that upregulate the expression of anti-pathogen genes and activate the adaptive immune system [1,2,3,4,5]. 

Several PRRs have been implicated in the antiviral innate response to murine retroviruses. This is likely due to PAMPs that are generated at different times and places during the retroviral replication cycle. For example, retroviral replication produces different forms of nucleic acids, such as single-stranded (ss) RNA or DNA, RNA:DNA hybrids, and double-stranded (ds) RNA or DNA. These molecules are found in the cytosol, in subcellular compartments, and in the nucleus during replication and within endosomes when virus particles are engulfed [6] (Figure 1). Additionally, retroviral virions and proteins are recognized by PRRs [7]. Thus, host cells have developed multiple mechanisms to detect retroviral PAMPs and thereby control or eliminate retrovirus infection.

## 2. Retroviruses Generate PAMPS during Replication

Retroviruses are enveloped, linear, nonsegmented ssRNA viruses that package two copies of the viral RNA genome [6]. The Retroviridae family is made up of a diverse group of viral subfamilies and genera; however, their common defining feature is the ability to replicate by reverse transcription. There are two major subfamilies of Retroviridae: Orthoretrovirinae and Spumaretrovirinae. The Orthoretrovirinae subfamily is further divided into six genera: *Alpharetrovirus*, *Betaretrovirus*, *Deltaretrovirus*, *Epsilonretrovirus*, *Gammaretrovirus,* and *Lentivirus*. Infectious murine retroviruses belonging to only two genera have been identified: the *Betaretrovirus* mouse mammary tumor virus (MMTV) and the *Gammaretrovirus* family of murine leukemia viruses (MLV).

Retroviruses encode *gag*, *pol,* and *env* genes. The Gag polyprotein is processed to generate the matrix, capsid and nucleocapsid structural proteins, and the viral protease. The reverse transcriptase (RT), which has both polymerase and RNaseH activity, and integrase (IN) enzymes are derived from *pol* [6]. The *env* gene encodes the Env polyprotein that is cleaved by host proteases called furins to yield the surface glycoprotein (SU) and transmembrane (TM) proteins. SU and TM are found on the virion surface and are needed for interaction with specific cell surface receptors. After interacting with a cell surface entry receptor, SU and TM undergo conformational changes which result in fusion of the viral and target cell membranes and release of viral capsids into the cytoplasm. The different strains of MLV use a variety of host multiple transmembrane-spanning proteins as entry receptors [8]. MLVs, like most retroviruses, enter cells by direct fusion with the cell membrane or after endocytosis into pH-neutral compartments [9,10,11]. In contrast, MMTV depends on acidic endosomes for fusion of viral and cell membranes after binding transferrin receptor 1, its entry receptor [12]. It has also been reported that in some cell types, MLV Env requires cleavage by cellular cathepsins located in acidic endosomes [13].

Murine retrovirus reverse transcription likely begins in the host cell cytosol upon release of capsid into the cytoplasm [6]. Minus-strand DNA synthesis is initiated by a host cell-derived tRNA primer packaged with the viral genomic RNA through its annealing to the primer binding site (PBS) found in the U5 region at the 5′ end of the RNA (Figure 1). RT-mediated DNA synthesis proceeds to the 5′ end of the genomic RNA, creating a short DNA intermediate called minus-strand strong-stop DNA (-sssDNA) annealed to the viral RNA. Viruses with mutations in RNaseH, which preferentially degrades RNA in RNA:DNA hybrids, do not proceed beyond this step since degradation of the RNA template is essential for -sssDNA transfer to the 3′ Terminal Repeat (R) region where DNA synthesis proceeds [14,15]. The transfer is facilitated by identical R regions at the 5′ and 3′ ends of the RNA. As synthesis continues, RNAseH degrades the viral RNA until it reaches a purine-rich, RNaseH-resistant section called the polypurine tract (PPT) [6,16]. Undigested PPT RNA mediates priming and synthesis of plus-strand DNA. Synthesis of plus-strand DNA proceeds until it reaches the tRNA primer, generating plus-strand strong-stop DNA (+sssDNA). Strand transfer then occurs as the tRNA is digested by RNaseH, and the PBS in +sssDNA anneals to the complementary PBS at the 3′ end of the minus-strand DNA. This allows the plus and minus strands to act as templates for each other to complete DNA synthesis, resulting in linear dsDNA. Thus, ssRNA, ssDNA, dsDNA, and RNA:DNA hybrids generated during the different steps of DNA synthesis can potentially be recognized as foreign by PRRs.

The newly synthesized viral dsDNA along with IN and other viral proteins necessary for integration are contained within the viral preintegration complex (PIC) that enters the nucleus [6]. Host cell mitosis is required for nuclear entry of the MLV and MMTV PICs because these viruses lack proteins to transport the PIC across the nuclear membrane [17,18]. After entry, IN, in concert with host proteins, and in the case of MLV, the viral p12 protein, tethers and joins the ends of the viral DNA to the host DNA to yield a provirus [19]. The provirus is then transcribed by host cell RNA polymerase II to generate the protein-coding mRNAs and the RNA genomes for packaging and virion production. Viral glycoproteins travel through the Golgi apparatus to the cell surface and arrive at the budding site where Env is inserted in the cell membrane. Gag assembles new virions by directing packaging of viral polyproteins and two molecules of dimerized plus-strand viral RNA into virions, either at the cell surface (MLV) or from cytoplasmic structures that traffic to the membrane (MMTV); in both cases, viral capsids then bud from the cell surface [6]. For both, newly synthesized viral mRNAs are a potential source of ligands for PRRs [20,21,22]. 

## 3. Murine Retroviruses and TLRs

The TLRs are an evolutionarily conserved group of type I transmembrane proteins that play important roles in fighting infection by many different pathogens [1,5]. Each TLR has evolved to recognize specific PAMPs; this ensures an innate immune response to different pathogens. TLRs contain a ligand-binding domain, located either extracellularly (TLRs 1, 2, 3, 4, 5, 6, and 11) or within cytoplasmic compartments such as endosomes and lysosomes (TLRs 3, 7, 8, and 9) and a cytoplasmic signaling domain called the Toll/IL-1R homology (TIR) domain. TLRs bind ligands, homo- or hetero-dimerize and then form homotypic interactions with a number of different cytoplasmic TIR-domain-containing adaptor proteins, including myeloid differentiation primary response 88 (MyD88), TIR-domain-containing adaptor protein-inducing IFN-β (TRIF)/TIR-domain-containing molecule 1 (TICAM1), TRIF-related adaptor molecule (TRAM), or TIR-associated protein (TIRAP)/MyD88-adaptor-like (MAL). The TLR-TIR-domain adaptor interaction triggers a signaling cascade culminating in activation of nuclear factor kappa-light-chain-enhancer of activated B cells (NF-κB) pathway or TANK-binding kinase 1 (TBK1) and interferon regulatory transcription factors (IRF) 3 or IRF7, thereby inducing production of proinflammatory cytokines and type I IFNs (Figure 2).

TLRs act as PRRs for a wide variety of PAMPs. For example, while TLR4 is the PRR for lipopolysaccharide (LPS) from Gram-negative bacteria, it also recognizes components of certain parasites and fungi, as well as viral and host proteins [1,5,23]. TLR2 forms heterodimers with TLR1 or TLR6 which recognize structurally different bacterial lipopeptides, TLR5 binds the flagellin protein of bacterial flagella, and TLR11 is the PRR for the protozoan profilin-like protein [1,5,24]. TLRs 2, 3, 4, 7, 8, and 9 also detect viral ligands. TLR2 interacts with viral proteins, largely as TLR2/6 heterodimers, and mediates cytokine production during lymphocytic choriomeningitis virus, New World arenavirus, measles virus, respiratory syncytial virus, and herpes simplex virus (HSV)-1 infections [25,26,27,28,29]. TLR3, found both on the cell surface and in endosomes, recognizes dsRNA, activates NF-κB, and generates in vivo or ex vivo immune responses during West Nile virus (WNV), influenza A virus (IAV), and herpesvirus infections [30,31,32,33,34]. TLR7 recognizes viral ssRNA and induces production of type I IFNs and proinflammatory cytokines during infection by IAV, vesicular stomatitis virus, and WNV and in response to transfected human immunodeficiency virus (HIV)-1 ssRNA [35,36,37,38]. The ligand for TLR9 is unmethylated CpG DNA, common to bacterial and viral DNA. Hence, TLR9 is important for IFN or cytokine production during herpesvirus, adenovirus, and poxvirus infections [39,40,41].

Several TLRs play a role in mediating immune responses during murine retrovirus infections. During early acute infection by intravenous inoculation with Friend virus (FV), which is a complex composed of Friend murine leukemia (F-MLV) helper virus and polycythemia-inducing spleen focus-forming virus, mice deficient in the ssRNA sensor TLR7 had higher levels of infectious virus in plasma which persisted until 14 days post infection (dpi), unlike wild-type (WT) controls, which had undetectable viremia at this time. Failure to control virus replication at early time points in TLR7-deficient mice was attributed to inability to mount effective IgM and IL-10 responses, both of which are enhanced by TLR7 signaling [42]. Deletion of the TLR7 adaptor, MyD88, also resulted in higher levels of FV infection in knockout (KO) mice compared with heterozygous controls, up to at least 16 weeks post-infection, and MyD88 KO mice were unable to generate FV-specific IgG responses. FV-infected mice with B cell-specific deletion of MyD88 also had much higher levels of virus in their spleens and significantly reduced FV-specific IgG in their serum compared with WT controls. TLR7 was required for this antibody response and for the development of germinal center B cells critical for appropriate antibody responses [43,44]. Similarly, both MyD88 and TLR7 KO mice were unable to mount virus-specific antibody responses after intraperitoneal injection of MMTV, and MyD88 KO splenocytes did not secrete proinflammatory cytokines in response to ex vivo exposure to MMTV. TLR7 KO mice infected with Rauscher-like MLV also failed to control virus and generate virus-specific antibodies [45]. Together, these data strongly suggest that detection of murine retroviruses and the subsequent adaptive immune responses are TLR7-MyD88-dependent. 

Conversely, a more recent study suggested that TLR7 signaling exacerbated early F-MLV infection and spread in the popliteal lymph nodes (pLN) of mice after subcutaneous inoculation of virus. TLR7 and type I IFN signaling activated B-1 cells, a highly susceptible population of cells within the pLN, and made them more susceptible to F-MLV infection. B-1 cells then spread virus to other B cell populations within the lymph node [46]. It is possible that the route of infection, virus dosage, or length of time after infection alters the ultimate outcome of infection. It is also possible that TLR7-mediated signaling creates a pool of actively dividing cells that are highly susceptible to MLV at early times post-infection, but in the long term, TLR7 signaling is needed to generate the innate immune response leading to adaptive immunity and infection control. 

In vivo studies have also demonstrated a role for TLR3 in murine retrovirus sensing. During acute FV infection, TLR3-deficient mice had increased viremia compared with WT mice. TLR3 depletion diminished expression of type I IFNs and interferon-stimulated genes (ISGs), the numbers of activated dendritic cells (DCs), and the cytotoxicity of natural killer cells and CD8^+^ T cells, all of which are important to the antiviral response [47]. Further support that activation of the TLR3 pathway can control retroviral infection comes from studies where polyI:C was used to stimulate TLR3 during FV infection in mice [48]. TLR3 stimulation increased type I IFN levels and CD4^+^ and CD8^+^ T cell responses and reduced viral loads, splenomegaly, and ultimately the development of leukemia in mice. Although these responses were dependent on TLR3, the polyI:C-treated mice were not able to completely control infection, so additional mechanisms must be necessary to clear virus. TLR3 likely detects endosomal dsRNA forms of MLV that occur when ssRNAs form secondary structures or when the two genomic ssRNAs form dimers.

Still others have identified TLR9 as a potential PRR for retroviral RNA:DNA hybrids in conventional and plasmacytoid DCs (pDCs). Although synthetic RNA:DNA hybrids activated TLR9-mediated signaling and RNA:DNA hybrids were detected in endosomes of cells persistently infected with Moloney MLV (M-MLV), the latter were not tested for their ability to activate TLR9-mediated signaling [49]. However, endosomal HIV-1 RNA induced IFN-alpha production and activation of pDCs, likely via TLR7 [50].

TLR 7 also plays a role in controlling spontaneous endogenous retrovirus (ERV) activation and tumor formation in mice. Specifically, TLR7 KO and MyD88 KO mice had dramatic increases in endogenous MLV expression; this also occurred in B cell-deficient mice and recombination activating 1 gene (*Rag1*) KO mice, which lack B and T cells, suggesting a role for TLR7-MyD88-mediated B cell responses in controlling the emergence of ERVs [51]. ERV reactivation resulted in retrovirus-induced lymphomas in the Rag1 KO mice through recombination between two nonfunctional ERVs to generate an infectious and pathogenic virus. Interestingly, the emergence of ERVs varied depending on where the TLR7 or MyD88 KO mice were housed, suggesting an environmental role for diet or microbiome in the activation of ERV expression.

TLR3, TLR7, and TLR9 were also implicated in the development of T cell acute lymphoblastic lymphoma (T-ALL) associated with ERV reactivation and reintegration in triple KO mice [52]. As in the TLR7/MyD88/Rag1 study, increased ERV levels and budding retroviruses depended on loss of TLR7. However, T-ALL developed only in conjunction with loss of TLR3 and 9 in aged triple KO mice. TLR3, TLR9 and TLR3/9 KOs all produced wild-type levels of ERV-specific antibodies when challenged with purified ERVs, whereas TLR7, TLR3/7, or TLR 7/9 double or TLR3/7/9 triple KO mice were unable to produce ERV-specific antibodies. These results suggest that ERV replication intermediates generated during ERV reactivation are detected, that TLR7 is important for generating a protective antibody response, and moreover, that TLR3 and TLR9 modulate this response, leading to tumor rejection.

In addition to the nucleic acid-sensing TLRs, TLR4 has been implicated in murine retrovirus sensing. The milk-borne retrovirus MMTV associates with the microbiota, specifically by binding LPS from intestinal bacteria, and activates TLR4- and MyD88-dependent signaling to induce expression of the immunosuppressive cytokine, IL-10; this in turn facilitates viral persistence and transmission via milk to pups [53]. To enable interactions with LPS and subsequent TLR4 activation and viral transmission, MMTV incorporates LPS-binding proteins TLR4 and CD14 into its membrane. As MD-2 directly binds LPS and forms a complex with TLR4, MD-2 was required on virion particles for TLR4 activation [54] (Figure 2).

MMTV interacts with TLR4 on the cell surface [55]. One as-of-yet unsolved aspect of the role of TLR3, TLR7, or TLR9 in detecting retroviruses is that their ligand-binding domains are located within endosomes or on the cell surface, yet release of viral RNA and reverse transcription occurs within the cytoplasm upon infection. This suggests that these nucleic acid-sensing TLRs are detecting defective retroviral particles, virus-infected cells engulfed by sentinel cells, or virions endocytosed as part of immune complexes. Indeed, TLR7 is known to encounter self RNAs that enter endosomes as RNA-autoantigen complexes via the B cell receptor or as RNA-immune complexes that are endocytosed via the Fc receptor [56,57].

## 4. ALRS, cGAS and Other Sensors

During infection, nucleic acids from viruses and bacteria found in the host cell cytosol or nucleus likely escape detection by TLRs, which are expressed on the cell surface or in endosomal compartments. Sensors such as RLRs, ALRs, and cGAS are responsible for the innate immune response to cytosolic and nuclear nucleic acids produced upon infection or in actively infected cells [58,59,60]. RLRs are not known to be involved in the antiviral response to mouse retroviruses and will not be further discussed [21]. cGAS is an enzyme that upon binding to dsDNA catalyzes synthesis of cyclic guanosine monophosphate–adenosine monophosphate (cGAMP), which directly binds to and activates stimulator of interferon genes (STING) [60] (Figure 2). The ALRs, also called PYHIN proteins, contain an N-terminal pyrin domain (PYD) and one or two C-terminal HIN domains [4]. The PYD domain is involved in protein–protein interactions with other PYD-containing proteins, while the HIN domain binds DNA. In contrast to the four *ALR* genes found in humans, there are 12–13 mouse *Alr*s, depending on the inbred mouse strain, encoded at a single locus on mouse chromosome 1 [58,61,62]. Several other sensors, including members of the DEAD box helicase and zinc-finger families, such as DEAD-box helicase 41 (DDX41) and zinc-finger antiviral protein (ZAP), respectively, have been implicated in retrovirus nucleic acid recognition [14,21,63]. 

Nucleic acid sensing by cGAS and DDX41, as well as several ALRs, activates the STING-dependent IFN induction pathway or the formation of inflammasomes and inflammatory cell death [3,4]. In contrast to cGAS, after DDX41 and ALRs detect nucleic acids, they likely directly interact with STING [58,61,63,64] (Figure 2). STING activation induces its dimerization, association with TBK1, and translocation from the endoplasmic reticulum (ER) to the Golgi apparatus. TBK1 then phosphorylates the transcription factors IRF3 and NF-κB, which translocate to the nucleus, leading to the production of type I IFNs and proinflammatory cytokines [65]. In addition to activating STING-dependent signaling, the ALRs IFI16 and AIM2 can form inflammasomes in human cells. They bind pathogen DNA at their HIN domains, which frees their PYDs to form homotypic interactions with the PYD of apoptosis-associated speck-like proteins containing a carboxy-terminal CARD (ASC). The caspase activation and recruitment domain (CARD) of ASC interacts with the CARD of pro-caspase-1, inducing cleavage and activation of pro-IL-1β, pro-IL-18, and gasdermin-D, resulting in inflammatory cell death called pyroptosis [4,66]. IFI16 detects abortive HIV-1 reverse transcripts in CD4 T cells, leading to pyroptosis; however, rather than protecting the host, this mechanism depletes CD4 T cells and contributes to progression to AIDS [67]. However, whether any of the murine ALRs aside from AIM2 play a direct role in inflammasome formation or if murine retroviruses activate the inflammasome has not been studied. 

Several studies have demonstrated the importance of cGAS and some of the mouse ALRs during the innate immune response to viral and bacterial infections. cGAS is important for IFN-β production in response to HSV-1 and *Francisella novicidia* infection [68,69]. The ALRs IFI204 in mouse cells and IFI16 in human cells are required for detecting transfected HSV-1 and vaccinia virus DNA, and IFI204 is required for production of type I IFNs and other cytokines during HSV-1, *F. novicida,* and *Mycobacterium bovis* infections [64,68,70]. Moreover, the release of mitochondrial DNA into the cytoplasm during infection by dengue virus, a flavivirus, leads to cGAS activation [71]. 

As reverse transcription occurs in the cytosol, these sensors also are important for innate immune responses during retroviral infections. Prior to the identification of cGAS and ALRs as retroviral sensors, it was demonstrated that sensing of reverse transcripts occurred because depletion of three prime repair exonuclease 1 (TREX1), a cytosolic ssDNA exonuclease that binds and degrades DNA, led to enhanced cytokine responses to HIV infection [72]. TREX1 degradation of reverse transcripts therefore dampens potential immune responses to retroviruses. However, it does not apparently target DNA destined for the PIC, since loss of TREX1 does not lead to increased proviral integration of HIV-1 in vitro or MLV in vivo [73] (Aguilera and Ross, in preparation). It thus potentially functions to promote retrovirus infection by degrading abortive reverse transcription products that would otherwise induce immune responses without diminishing the dsDNA viral genome destined for chromosomal integration.

The nucleic acid sensor cGAS is essential for IFN-β and CXCL10 induction in the in vitro response to MLV infection in TREX1-depleted L929 cells and murine embryonic fibroblasts (MEFs) [74]. This STING-dependent response was also observed during infection with non-murine retroviruses such as HIV and demonstrated that cGAS is a common sensor of retroviral DNA. In addition to cGAS, DDX41 and the ALR IFI203 were identified as STING-dependent sensors of TREX1-sensitive MLV reverse transcripts in infected murine macrophages and DCs. IFI203 and DDX41 interacted with each other and with MLV reverse transcripts in the cytosol [63]. Additional studies determined that DDX41 binds the initial products of reverse transcription, MLV RNA:DNA hybrids, while cGAS binds dsDNA generated at a later step, and that both functioned synergistically to induce type I IFNs via STING. Both DDX41 and cGAS were required for sensing and control of MLV in vivo; this control required DDX41 expression in DCs but not macrophages [14]. 

Recent studies have shown that ALRs such as IFI16 in humans and IFI204 in mice also function as transcriptional repressors of virus gene expression. For example, IFI16 has been shown to decrease HSV-1 and HIV-1 gene expression in the nucleus [75,76]. Similarly, the murine ALR IFI204 decreased HIV-1 gene expression and shedding of RT-containing MLV virions when ectopically overexpressed in human cells, and complete knockout of the entire ALR locus in mice led to increases in plasma viremia and smaller, yet significant, increases in the numbers of infected cells during acute FV infection [76]. This is interesting, in light of a report that this same deletion of the ALR locus did not alter the response to transfected DNA or human cytomegalovirus in primary cells derived from the same KO mice [77]. 

The ALRs AIM2 and IFI205 along with cGAS also play a role in sensing endogenous retroelement DNA [61,78]. In TREX1 KO mice, DNA accumulates in tissues and is detected by sensors which signal through the cGAS–STING pathway, activating immune cells and causing severe inflammatory tissue damage, particularly in the heart [78,79]. Mutations in human *TREX1* are linked to inflammatory and autoimmune diseases such as Aicardi–Goutières syndrome [80]. In the absence of TREX1, IFI205 in mouse macrophages and cGAS in mouse cells or heart tissue can detect cytosolic retroelement or self-DNA and stimulate type I IFNs and ISGs [61,78]. However, when AIM2 is expressed, IFI205 is blocked from interacting with STING and unable to activate immune responses to retroelement DNA [61]. This, along with TREX1, likely helps to prevent overactivation of the innate immune response and autoimmunity. IFI205 has also been shown to activate transcription of *Asc*, and IFI205 depletion results in decreased inflammasome activation in response to transfected dsDNA [81]. Whether endogenous retroviruses also increase ASC levels via IFI205 has not been tested. Interestingly, *Alr* genes have been implicated in autoimmune diseases such as lupus in mice [82]. 

Another antiviral host factor that diminishes the host response to retroviruses is apolipoprotein B editing, catalytic subunit 3 (APOBEC3). While the number of *APOBEC3* genes varies among mammalian species, mice encode a single *Apobec3* gene (mA3) [83]. APOBEC3 proteins are cytidine deaminases which are packaged into virions and induce G-to-A hypermutations in DNA. Human APOBEC3 proteins act on minus-strand DNA during RT via this mechanism to restrict retrovirus replication [84]. However, APOBEC3 proteins also inhibit reverse transcription, likely by binding to RT [85] (Figure 2). Interestingly, mouse APOBEC3 largely blocks infection by murine retroviruses like MMTV and MLV by this mechanism [86,87]. While this limits productive infection, it also has the potential to reduce the level of ligands that can be detected by nucleic acid sensors. Indeed, APOBEC3 KO cells and mice have higher interferon and cytokine responses to MLV infection than WT mice [63]. 

ZAP, another cytosolic sensor, was identified because of its ability to deplete cytosolic M-MLV mRNAs and inhibit retroviral gene expression in infected cells [22,88]. M-MLV also replicates to higher levels in ZAP KO MEFs compared with WT MEFs [21]. When ZAP is ectopically expressed, it localizes to RNA-containing stress granules [89]. It then recruits MLV RNA by binding to the U3 region via its zinc-finger domains and to exosome components that degrade the MLV transcripts [21,88]. Thus, rather than initiating a signaling cascade that signals to other cells to arm themselves against infection, these data suggest that ZAP behaves as a cell-intrinsic antiviral sensor that causes the degradation of retroviral cytosolic RNA destined for translation or packaging into virions. 

## 5. Viral Proteins that Block Host Sensors

Different murine retroviruses use a variety of mechanisms to protect against host sensors. For example, the capsids of MLV and other retroviruses act as protection against host nucleic acid sensors. Reverse transcription requires disassembly or loosening of the virion structure. Mutations in a *gag*-encoded protein found in *Gammaretroviruses* called glycoGag cause the capsid to fall apart more easily, resulting in increased reverse transcription and more ready access by host nucleic acid sensors to viral nucleic acids [63,87,90]. In contrast, capsid-stabilizing mutations do not undergo reverse transcription, and although this has not been tested, these viruses likely would not be detected by host sensors [91]. MMTV RT, on the other hand, carries out very rapid reverse transcription compared with other retroviral RTs [92]. This may limit the time that the different forms of nucleic acid are accessible to host sensors. The MLV capsid also impedes access of APOBEC3 to the reverse transcription complex, since glycoGag mutant viruses are more sensitive to APOBEC3-mediated inhibition of reverse transcription than are WT viruses [63,87]. Furthermore, MLVs encode a protein called P50 made from an alternatively spliced transcript, which interacts with APOBEC3 to block its packaging into new virions and to block APOBEC3-dependent restriction of MLVs both in vitro and in vivo [93].

Capsid sequence or structure may also limit the ability of host protein sensors, such as Tripartite motif (TRIM5α), to detect MLV capsids [94,95]. Although the retroviral capsid can protect against APOBEC-mediated restriction of viral reverse transcripts, the capsid lattice structure can also act as a PAMP and mediate innate immune signaling. TRIM5α protein, an E3 ubiquitin ligase, binds to the capsids of HIV-1 and other retroviruses, including N-tropic MLV, whose CA sequence differs from M-MLV, a B-tropic virus (N and B refer to the mouse strains susceptible to these viruses). After binding capsid, TRIM5α autoubiquitinates and induces transcription of inflammatory genes via NF-κB and MAPK signaling [96,97,98]. In addition to its role as a PRR, TRIM5α restricts HIV-1 and N-tropic MLV by degrading or disassembling viral cores, thereby reducing both reverse transcription and integration products [99]. Humans have a single *TRIM5α* gene, while mice have eight or more *Trim5*-like genes due to expansion of the locus [100]. A recent study suggested that mouse TRIM12c is a homolog to human TRIM5α. TRIM12c had ubiquitin-ligase activity, induced IFN-β and NF-κB promoter activity when overexpressed, and restricted replication of mouse stem cell virus, a MLV-based retroviral vector; however, whether this was due to capsid recognition was not determined [101].

## 6. Conclusions

Immune responses to retroviruses and ERVs are mediated by TLRs, ALRs, cGAS, and other cytosolic sensors. This wide variety of receptors and the intricate signaling pathways which they activate are necessary to facilitate targeted immune responses as endogenous and exogenous retroviruses produce multiple ligands and replicate in many different cellular compartments, cells, and tissues. These sensors and signaling pathways mediate B and T cell activity, activation of type I IFNs, cytokines, and other antiviral genes, and stimulate antibody responses to effectively reduce viral loads and protect the host. When appropriate, these PRRs also negatively regulate inflammatory responses to reactivated ERVs to prevent autoimmunity and tumor formation. Although these highly tuned signaling pathways have evolved to protect the host, retroviruses have also evolved to counteract or evade these protective host responses, facilitating their persistence and pathogenesis. 

The use of naturally infectious retroviruses in genetically modified mice lacking different PRRs, nucleic acid sensors, and downstream effector molecules has provided much information about their role in immune responses and control of infection. Yet to be determined are (1) how binding of ALRs and DDX41 to STING activates it; (2) how endosomal PRRs recognize viral PAMPs during natural infection; (3) why mice have an expanded ALR locus and whether pathogens like retroviruses, or endogenous retroviruses contributed to this expansion; and (4) given the persistence of these retroviruses in mice for millions of years, whether there are as-of-yet undiscovered viral proteins that block sensor recognition?

## Figures and Tables

**Figure 1 viruses-12-00836-f001:**
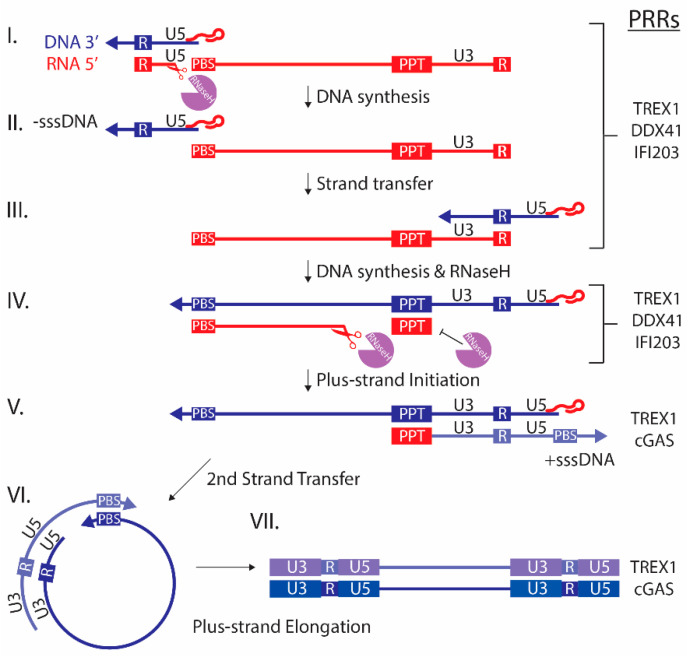
Retroviral reverse transcription. (**I**) Cellular tRNA binds to the primer binding site (PBS) in the Unique 5′ Region (U5) region of the RNA, initiating minus-strand DNA synthesis. (**II**) Synthesis proceeds to the 5′ end of the RNA, generating minus-strand strong-stop DNA (-sssDNA). (**III**,**IV**) RNaseH degrades the RNA template, enabling -sssDNA transfer to the 3′ R region (identical to the 5′ R region) where DNA synthesis and RNaseH digestion proceed. (**V**) RNaseH is unable to digest the polypurine tract (PPT), leaving a segment of RNA which primes synthesis of plus-strand DNA. Synthesis proceeds until reaching the tRNA primer, creating plus-strand strong-stop DNA (+sssDNA). (**VI**) A second strand transfer occurs, where the +sssDNA PBS anneals to the 3’ minus-strand PBS. (**VII**) The two strands act as templates for each other to complete dsDNA synthesis, generating identical long terminal repeats (LTR) at both ends of the dsDNA, each containing U3-R-U5 sequences. Shown to the right are the PRRs that recognize the nucleic acids generated at the different reverse transcription steps. Red, RNA; blue, minus-strand DNA; purple, plus-strand DNA.

**Figure 2 viruses-12-00836-f002:**
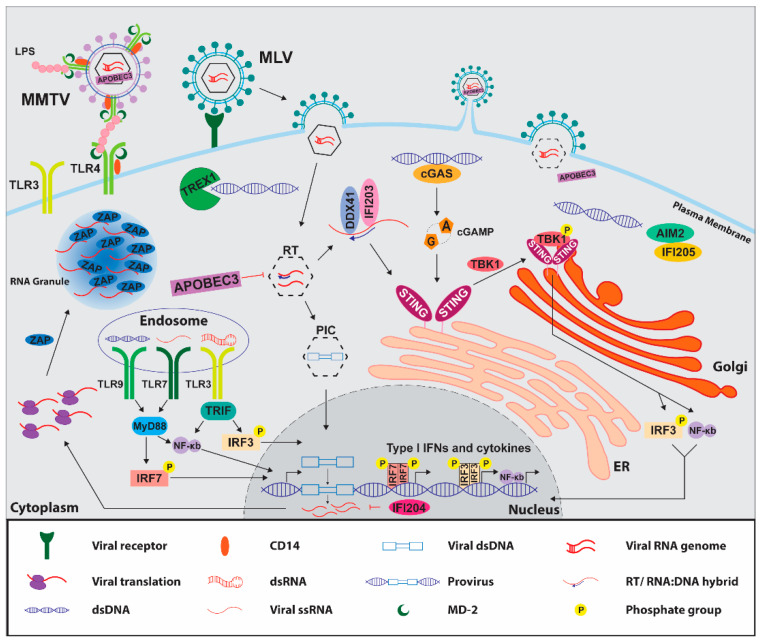
Retroviral replication and innate immune sensing. Murine retroviruses bind cell surface entry receptors, fuse with the cell membrane, either at the plasma membrane (MLV) or in endosomes (MMTV), and release their viral capsids into the cell cytoplasm, where reverse transcription takes place. Cytosolic sensors such as DDX41, IFI203, and cGAS detect viral reverse transcripts and upregulate transcription of type I IFNs and cytokines through STING-dependent signaling, which results in the phosphorylation of TBK1 and the transcription factors IRF3 and IRF7, or in NF-κB activation. TREX1 degrades reverse transcripts and endogenous retroelement DNA and hinders immune responses to these ligands. IFI205 stimulates an immune response to endogenous retroelement DNA; however, AIM2 blocks this STING-dependent immune response by sequestering IFI205 from STING. APOBEC3 blocks murine retroviral reverse transcription in the cytosol and diminishes innate immune responses. However, some fraction of newly synthesized viral DNA enters the nucleus via the viral PIC, and viral DNA integrates into the host cell DNA. The integrated proviral DNA is transcribed to generate viral mRNAs and new copies of the viral genome for packaging. Sensors such as IFI204, which can localize to the nucleus, are capable of inhibiting retroviral gene transcription. The cytosolic RNA sensor, ZAP, recruits viral mRNAs and exosome components to RNA-containing stress granules, resulting in degradation of retroviral mRNAs. Viral mRNAs that avoid restriction enter the cytoplasm to begin translation of viral proteins. TLRs also detect viral ssRNA (TLR7), dsRNA (TLR3), and dsDNA (TLR9) in endosomes and initiate cytokine responses via the adaptors MyD88 and TRIF. Packaging of new virions occurs at the cell membrane (MLV) or in structures in the cytoplasm which are later transported to the membrane (MMTV). MMTV incorporates LPS-binding proteins such as MD-2, TLR4, and CD14 into its membrane, which subsequently bind to bacterial LPS and facilitate virus transmission to pups via milk.

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
