# Peer review of "Insights into Sensing of Murine Retroviruses"

_viruses, 2020, doi:10.3390/v12080836_

Round 1

Reviewer 1 Report

The manuscript entitled “Insights into Sensing of Murine Retroviruses” by Moran and Ross documents the role of various arms of host innate immune system that may restrict infection of murine retroviruses.

The review provides a detail of various sensors that recognize the viral pathogen to mount an antiviral response. The review, however, lacks the information on the specific proteins/ genes of various murine retroviruses that are involved in modulating the antiviral response and may restrict retrovirus infection. This information is important and should be discussed in the context of different pattern recognition receptors and how it impacts the virus infection in vitro and in vivo.

Proof-reading is needed for minor concerns such as removing the para indent at line 142

Reviewer 2 Report

In this review, the authors summarize what is known about murine innate immune detection and response to murine retroviruses. This review is important, as mouse models are an essential tool to understand host immune-viral interactions in vivo. Furthermore, the current SARS-CoV-2 pandemic has highlighted the importance of understanding the immune response to viral infection and has triggered a renewed interest by scientists into understanding the innate immune response and how viruses can circumvent surveillance. The authors provide a nice summary of how MLV, MMTV, and ERVs are detected and controlled in the murine system.

minor critiques:

  1. Figure 1 can do with I) II) III) etc for the different steps of retroviral reverse transcription. A nice addition to this figure would be to include on the right or left the PRRs or nucleic acid sensors implicated in detecting each of these nucleic acid products.
  2. Figure 2 does not show AIM2 yet it is discussed in the text. 
  3. Figure 2 - The figure legends do not discuss the role of phosphorylation of transcription factors downstream of PRR. The figure legend also does not mention MD-2 and the role it plays in viral entry. I also can't find mention of MD-2 in the text. 
  4. Figure 2 - can you define either in the figure or the text what an RNA granule is? Stress granules are mentioned near the end but it is not clear if these are the same as RNA granules. How are they defined microscopically and biologically?
  5. Figure 2 - TLR adaptors, TREK1, and TBK1 are not mentioned in the figure legends. 
  6. Figure 2 - can you mention which TLRs indicated in the figure bind ssRNA, dsRNA, and dsDNA? 
  7. In the discussion of ALRs (section 4), it is mentioned that ALRs have PYD and HIN domains. Can you add what is known about the order of events for DNA binding by HIN domains and multimerization by PYD domains and how this triggers downstream inflammatory signaling? Does this sensing result in inflammasome formation and caspase-1 activity?
  8. In the paragraph concerning pDCs and TLR9, it would be beneficial to mention that human pDCs detect HIV via TLR7 (PMID: 16224540).
  9. Can you add a small section at the end on what are the key questions left in the field? 
